# Platelet-to-White Blood Cell Ratio: A Feasible Biomarker for Pyogenic Liver Abscess

**DOI:** 10.3390/diagnostics12102556

**Published:** 2022-10-21

**Authors:** Dong-Gyun Ko, Ji-Won Park, Jung-Hee Kim, Jang-Han Jung, Hyoung-Su Kim, Ki-Tae Suk, Myoung-Kuk Jang, Sang-Hoon Park, Myung-Seok Lee, Dong-Joon Kim, Sung-Eun Kim

**Affiliations:** 1Department of Physiology, College of Medicine, Hallym University, Chuncheon 24252, Korea; 2Department of Internal Medicine, Hallym University Sacred Heart Hospital, Medical Center, 22, Gwanpyeong-ro 170 Beon-gil, Anyang-si 14068, Korea; 3Institute for Liver and Digestive Diseases, Hallym University, Chuncheon 24252, Korea; 4Department of Internal Medicine, Dongtan Sacred Heart Hospital of Hallym University Medical Center, Hwaseong-si 18450, Korea; 5Department of Internal Medicine, Kangdong Sacred Heart Hospital of Hallym University Medical Center, 18, Cheonho-daero 173-gil, Seoul 05355, Korea; 6Department of Internal Medicine, Chuncheon Sacred Heart Hospital of Hallym University Medical Center, 77, Chuncheon-si 24253, Korea; 7Department of Internal Medicine, Kangnam Sacred Heart Hospital of Hallym University Medical Center, 1, Singil-ro, Seoul 07441, Korea

**Keywords:** platelet-to-white blood cell ratio, pyogenic liver abscess, pleural effusion

## Abstract

The platelet-to-white blood cell ratio (PWR) has been reported to predict the severity of patients with various diseases. However, no previous studies have assessed the use of the PWR as a prognostic marker for pyogenic liver abscesses (PLA). This observational retrospective study was performed between January 2008 and December 2017, including 833 patients with PLA from multiple centers. The enrolled patients, on average, had a PWR of 17.05, and 416 patients had a PWR lower than 17.05. A total of 260 patients (31.2%) with PLA showed complications of metastatic infection, pleural effusion and abscess rupture. A low PWR level was identified as a strong risk factor for metastatic infection and pleural effusion. The low PWR group also had a longer hospital stay. In the multivariate analysis, old age, anemia, albumin and CRP levels and unidentified pathogens were significant factors for low PWR levels. A low PWR, old age, male sex, abscess size, albumin, ALP and unidentified causative pathogens showed significant associations with a hospital stay longer than 28 days. As a result, PLA patients presenting with a low PWR were shown to have more complications and a poor prognosis. Considering its cost-effectiveness, PWR could be a novel biomarker used to predict a prognosis of PLA.

## 1. Introduction

Pyogenic liver abscesses (PLA) have a low and varying rate of incidence, depending on the countries in which it is measured, ranging from 1.1 to 4.1 per 100,000 in Western countries [1,2,3,4] to 5.6-17.6 per 100,000 in Eastern countries [5,6,7]. In recent studies, the mortality rate of patients with PLA ranged from 2.3% to 15% [1,7,8,9]. The risk factors for mortality, such as old age and immunosuppression, are well known. Patients with PLA presenting with multiple abscesses, hypoalbuminemia, leukocytosis, jaundice, bacteremia, related malignancies and any significant complications have a higher mortality rate [9,10]. Although the mortality of PLA has experienced a significant decline over the decades, it currently remains considerably high. In addition, recent studies have reported a rapid increase in the incidence of PLA in Asia, especially Korea [7,11]. This phenomenon is known to be related to an increase in the numbers of older patients and patients with underlying diseases such as cancer, cirrhosis and chronic renal failure [7,10] and increasing infections with not only *Klebsiella pneumoniae (K. pneumoniae*) [7,12] but also multidrug-resistant organisms [13]. In clinical settings, the initial recommended treatment for PLA is the use of empiric antibiotic regimens such as amoxicillin-clavulanic acid, third-generation cephalosporins combined with an aminoglycoside, or piperacillin/tazobactam [14]. However, Park et al. reported that multidrug-resistant-organism-related PLA accounts for 6.6% of all PLAs. Therefore, these pathogens may cause the initial empiric antibiotics used for PLA to fail, but the predictors of this failure have not yet been evaluated.

Recently, hematological markers for the prognosis of several diseases have become an area of focus in research. The neutrophil-to-lymphocyte ratio, monocyte-to-lymphocyte ratio, red cell distribution width and mean platelet volume have been suggested to be inflammatory markers with prognostic value in various diseases [15,16,17,18]. In particular, the platelet-to-white blood cell (WBC) ratio (PWR) is increasing in importance in diverse medical settings. As an economic yet powerful hematologic biomarker, PWR has been shown to be correlated with the prognosis of patients with conditions such as acute ischemic stroke [19], advanced ovarian cancer [20], liver failure [21], hepatitis B virus-associated liver cirrhosis [22], cirrhosis with acute decompensation [23], intrahepatic cholangiocarcinoma [24], and even trauma [25] and metabolic syndrome [26] based on the WBC counts measured in some studies. Among the studies, PWR has been shown to be correlated with the severity of systemic inflammation. Provoking leukocyte–platelet aggregates, in which leukocytes activate platelets, to increase inflammation increases the WBC counts and excessive consumption of platelets, leading to a lower PWR. A plausible hypothesis is that PWR is a biomarker for systemic inflammation, in which a severely low PWR would lead to immune disruption. Although PWR is considered an important biomarker of infectious diseases, few studies related to this topic have been published. In the present study, we aimed to investigate the role of PWR as a predictor of complications and the prognosis of patients with PLA. 

## 2. Materials and Methods

### 2.1. Patients and Data Collection

This observational retrospective study was performed between January 2008 and December 2017 at Hallym University Medical Center, which consists of 6 hospitals: Hallym University Sacred Heart Hospital, Hallym University Kangnam Sacred Heart Hospital, Hallym University Chuncheon Sacred Heart Hospital, Hallym University Hangang Sacred Heart Hospital, Hallym University Dongtan Sacred Heart Hospital and Kangdong Sacred Heart Hospital in Korea. The medical records of Hallym University Medical Center were systematically searched for code K75.0 of the International Classification of Disease, 10th Revision, to identify patients with PLA. Enrolled patients satisfied the following criteria: (1) compatible findings on ultrasound (US), computed tomography (CT) or magnetic resonance imaging (MRI) diagnosing PLA, (2) either positive culture results from blood culture or an abscess and (3) the resolution of symptoms after antibiotic therapy. Patients meeting the following criteria were excluded: (1) age < 18 years old and (2) diagnosis of parasitic/fungal/amoebic abscesses.

The medical records were systemically reviewed, including clinical information on sex, age and underlying and current medical conditions. Patients with clinical symptoms indicative of cholecystitis, cholangitis or documented bile duct disease were considered to have PLA secondary to biliary tract disease. Initial laboratory values were collected on the first day of hospital admission or within 24 h after the clinical diagnosis of PLA if PLA was not the initial cause of hospitalization. Blood cultures were performed for all patients. If patients were treated with the percutaneous drainage or aspiration of the abscess, pus cultures were performed with the collected material. The PWR was calculated by dividing two values from the laboratory results. The PWR cutoff value for the low PWR group was selected based on the median PWR values of the total number of patients with PLA included in the study. Anemia was defined as a hemoglobin (Hb) level less than 13 g/dl in adult males and less than 12 g/dl in adult females.

The requirement for informed consent was waived because this study was designed as a retrospective study, and the analysis used anonymous clinical data. This study was approved by the Ethics Committee of Hallym University Medical Center. The study was conducted with the approval of the Institutional Review Board of Hallym University Medical Center (2019-06-007-001). All methods were conducted in accordance with the relevant guidelines and regulations.

### 2.2. Treatment Protocol

The basis of treatment was antibiotic therapy and percutaneous drainage (PD) according to size and nature of PLA. All patients with PLA were treated promptly when PLA was suspected after at least one set of blood cultures were taken. Empiric antibiotic regimens usually consist of a third-generation cephalosporin, such as ceftriaxone, along with metronidazole. PD was performed when (1) the size of PLA was ≥ 5 cm, (2) the presence of hemodynamic instability was observed, (3) gas within the abscess cavity, regardless of size, was observed and (4) the failure of antibiotic therapy for PLA < 5 cm (5), a sign of impending rupture on radiologic imaging, was observed [27]. The antibiotic regimen was subsequently tailored to the culture and sensitivity results of the blood or pus. The recommended duration of intravenous antibiotic treatment is around 2 weeks before transitioning to oral antibiotics. The total duration of antibiotic treatment varied according to the clinical and radiological response of each PLA patient. Depending on the radiologic assessment of the size and status of residual PLA, the drainage tube was either repositioned, upsized or withdrawn. The drains were removed when we observed a decrease of <10 mL/day in the drain aspirates for at least 2 days or the resolution of PLA with clinical improvement, as evidenced by stable vital signs, including WBC and C-reactive protein (CRP) down-trending. Discharge criteria for patients who had received intravenous antibiotic treatment for around 2 weeks were defined as the time when (1) the patient’s vital signs were stable, (2) the inflammatory markers, such as the WBC count and CRP level, were improved and (3) the drain could be removed. Treatment outcomes were measured by the length of hospital stay and mortality. The length of hospitalization was 14 days and 28 days, respectively. All the patients were recommended to undergo repeat radiologic imaging at 2–3 week intervals to evaluate the treatment response. Antibiotic treatment was discontinued when there was a clinical resolution or near-complete to complete radiological resolution of PLA after a minimum duration of 6 weeks. 

### 2.3. Statistical Analysis

The statistical analysis was performed using R, a language and environment for statistical computing (R Foundation for Statistical Computing, Vienna, Austria) and SPSS version 24 software (IBM Co., Armonk, NY, USA). Quantitative variables are presented as the means ± standard deviations (SDs). The skewness of the value was calculated, and values > −2 and <2 were considered evenly distributed, and the mean value was used as a representative value. Groups with skewed values who did not satisfy this standard were considered skewed, and the median value was used to represent that group. Welch’s two-sample test and Student’s t-test were used to analyze continuous variables by calculating the statistical results of the F test to compare the differences between two variables, and Fisher’s exact or Pearson’s chi-square tests were used to compare categorical variables. Factors that were significant in the univariate analysis were entered into a stepwise multivariate analysis to determine the independent risk factors. Odds ratios and their 95% confidence intervals were calculated. All *p* values < 0.05 were considered statistically significant, and all *p* values were two-tailed. Mortality was assessed by analyzing the Kaplan–Meier curves.

## 3. Results

### 3.1. Baseline Characteristics

Among the total of 833 patients enrolled in the study, the median PWR of all patients with PLA was 17.05 (range: 0.12-237.40). Considering the left-skewed distribution of PWR (skewness 4.01), an adequate cutoff value for a low PWR was determined as the median value. The baseline characteristics of the enrolled patients are shown in Table 1. Four hundred and sixteen (49.9%) patients had PWRs lower than 17.05. Patients with a low PWR seemed to be older than those with a high PWR (64.1 ± 14.9 years vs. 60.4 ± 15.6 years; *p* < 0.001). The low PWR group had significantly higher WBC counts and lower platelet counts than the high PWR group. The CRP level was increased in the low PWR group (206.8 ± 91.3 mg/dL) compared to the high PWR group (161.9 ± 340.7 mg/dL) (*p* = 0.01). Patients with a low PWR showed an association with hypertension (HTN), but no significant association was observed with diabetes mellitus (DM). Interestingly, the Hb levels differed between the two groups, in which patients with a low PWR had higher levels of Hb than patients with a high PWR (12.5 ± 2.1 g/dL vs. 12.2 ± 2.00 g/dL; *p* = 0.01). Albumin levels were decreased in the low PWR group compared with the high PWR group (3.3 ± 0.6 g/dL vs. 3.5 ± 0.6 g/dL; *p* < 0.001). Aspartate transaminase (AST) and alkaline phosphatase (ALP) levels were not significantly different between the PWR groups, whereas alanine transaminase (ALT) levels were increased in the low PWR group (96.0 ± 115.9 IU/L) compared to the high PWR group (76.0 ± 126.5 IU/L; *p* = 0.02). The size and location of liver abscesses were analyzed for all patients who showed suspicious findings on radiologic exams. Of the 833 patients who had an abscess with an average size of 5.4 ± 2.8 cm, no significant difference in the abscess size was observed between the PWR groups (5.4 ± 2.6 cm vs. 5.3 ± 3.1 cm; *p* = 0.071). The location of the abscess was analyzed based on the lobes, and the most common site of occurrence was the right side, in 527 patients (63.3%), followed by the left (172 patients, 20.6%) and both lobes (134 patients, 16.1%). Similar tendencies were observed in both PWR groups, but the lower PWR group included a higher proportion of patients with abscesses on the left side than the higher PWR group (24.0% vs. 17.3%; *p* = 0.02) compared to the right side (59.6% vs. 66.9%; *p* = 0.03). The rate of occurrence in both lobes was not different between the PWR groups (16.3% vs. 15.8%, *p* = 0.84). One hundred and eighty-one patients (21.7%) with PLA experienced complications, including metastatic infections, such as septic endophthalmitis or pneumonia, pleural effusion and abscess rupture. The PWR of patients with complications was calculated as 18.6 ± 19.6 with a WBC count of 14.78 ± 7.04 × $10^{3}$ cells/μl. Among the patients with complications, pleural effusion was the most common, which occurred in 85 patients (47.0%). Septic endophthalmitis, which is known to be the most common metastatic infection, was present in 15 patients (8.3%), whereas cases of pneumonia were reported in 29 (16%), with the remaining 44 patients (24.3%) having metastasis. Abscess rupture was only observed in eight patients (4.4%). Patients with PLA presenting a low PWR showed a significantly greater rate of development of complications, including metastatic infection, than those with a high PWR (29.8% vs. 13.7%; *p* < 0.001, 50.8% vs. 43.9%; *p* = 0.0014), including septic endophthalmitis (9.7% vs. 5.3%, *p* = 0.02) and pneumonia (17.7% vs. 12.3%, *p* = 0.004). However, the opposite trend was observed for pleural effusion between the PWR groups (45.2% and 50.9%, respectively, *p* = 0.0019). The incidence of abscess rupture between the low and high PWR groups was not significantly different (4.0% vs. 5.3%, *p* = 0.48). Causative pathogens were identified in 622 of the total of 833 patients. *K. pneumoniae* was the predominant pathogen, accounting for 48% of all infections (400/833). Interestingly, *K. pneumoniae* was detected as the causative pathogen in 56.3% of patients with PLA presenting with a low PWR compared to 39.8% of patients with PLA presenting with a high PWR (234/416 vs. 166/417, *p* < 0.001). However, a significant difference in the identification of *Escherichia coli (E. coli)* was not observed between the low PWR and high PWR groups.

### 3.2. Analysis of Factors Related to a Low PWR in Patients with PLA

We performed univariate and multivariate logistic analyses to investigate the factors related to a low PWR in patients with PLA (Table 2). In the univariate analysis, an older age, HTN, location of the abscess, anemia, albumin levels, AST levels, ALT levels, CRP levels, *K. pneumoniae* infection and unidentified infection were significant factors contributing to the development of low PWR levels. In the multivariate analysis, an old age > 65 years, anemia, albumin and CRP levels and infection with unidentified pathogens were significant factors contributing to low PWR levels (odds ratio (OR) 1.918; 95% confidence interval [17] 1.330–2.780; *p* < 0.001, OR 0.423; 95% CI 0.288–0.615; *p* < 0.001, OR 0.617; 95% CI 0.447–0.844; *p* < 0.001, OR 1.0005; 95% CI 0.999–1.002; *p* < 0.001, OR 0.545; 95% CI 0.334–0.833; *p* = 0.014, respectively). However, no significant associations were observed between sex, DM, abscess location, AST levels, ALT levels, ALP levels, PT-INR, Cr levels, causative organisms of *K. pneumoniae* or *E. coli* and low PWR levels in patients with PLA.

### 3.3. Analysis of Risk Factors for Complications of PLA

We performed univariate and multivariate logistic regression analyses of patients with PLA to reveal the risk factors for the development of metastatic infection in detail (Table 3). In the univariate analysis, a low PWR, anemia, *K. pneumoniae* and unidentified infection were associated with the development of metastatic infection. In the multivariate analysis, a low PWR level was significantly associated with the development of metastatic infection (OR 1.953; 95% CI 1.163-3.364; *p* = 0.013). However, age, HTN, DM, the location and size of the abscess, levels of albumin, AST, ALT, ALP, PT-INR, Cr, and CRP and infectious pathogens did not show significant associations with the development of metastatic infection in patients with PLA.

Additionally, univariate and multivariate logistic analyses were performed to identify the risk factors for the development of pleural effusion (Table 4). In the univariate analysis, a low PWR, abscess size, anemia, albumin level, ALP level, *E. coli* infection and infection with an unidentified organism were associated with the development of pleural effusion. In the multivariate analysis, a low PWR, ALP level and infections with an unidentified organism were associated with pleural effusion (OR 2.007; 95% CI 1.204–3.419; *p* = 0.0087, OR 1.001; 95% CI 1.000-1.002; *p* = 0.006, OR 0.458; 95% CI 0.218–0.883; *p* = 0.027, respectively). Other factors, including age, HTN, DM, the location and size of abscess, anemia, levels of albumin, AST, ALT, PT-INR, Cr, and CRP and infections with *K. pneumoniae, E. coli* and other organisms were not correlated with the development of pleural effusion.

### 3.4. Comparison of the Mortality Rate and Hospital Stay between PWR Groups

The mortality rates of the low PWR group and high PWR group are compared in Figure 1. A total of 34 patients (4.1%) out of 833 patients enrolled in the study died during hospitalization. A significant difference in overall mortality was not observed between the low and high PWR groups (4.57% vs. 3.60%, *p* = 0.99). Because the survival probability of the total group was very high at all time intervals, we focused on the trends between the groups but not the actual probability. The low PWR group showed an acute decrease in the probability of survival at approximately day 60, with a survival probability of approximately 70% in the normal PWR group on day 87. This implies a trend of a decreasing survival probability in patients with a low PWR.

Patients were also stratified by the length of the hospital stay, and the results are shown in Figure 2. More patients with a low PWR required prolonged hospitalization compared to patients with a high PWR (hospital days (HD) 14 (80.5% vs. 63.1%, *p* = 0.001) and 28 (40% vs. 31.56%, *p* < 0.001)).

We conducted univariate and multivariate logistic regression analyses of the subgroups (HD 14 and 28, Table 5 and Table 6, respectively) to identify risk factors for prolonged hospitalization. In the univariate analysis, a low PWR, abscess size, anemia, albumin, ALP and CRP levels, *K. pneumoniae* infection and unidentified infection were associated with prolonged hospitalization for more than 14 days. In the multivariate analysis, a low PWR, abscess size, and levels of ALP and CRP showed correlations with prolonged hospitalization for more than 14 days (OR 1.573; 95% CI 1.053–2.357; *p* = 0.027, OR 1.094; 95% CI 1.011–1.186; *p* = 0.028, OR 1.002; 95% CI 1.0007–1.003, *p* = 0.002, OR 1.005; 95% CI 1.003–1.008; *p* < 0.001, respectively). The remaining factors did not significantly affect the number of HDs.

For patients who were hospitalized for more than 28 days, the univariate logistic analysis revealed that a low PWR, older age, male sex, abscess size, levels of albumin, ALT, and ALP, infection with *K. pneumoniae* and unidentified infection were associated with prolonged hospitalization. In the multivariate logistic analysis, a low PWR, older age, male sex, abscess size, levels of albumin and ALP and infection with unidentified causative pathogens showed correlations with an adverse outcome (OR 1.738; 95% CI 1.211–2.507; *p* = 0.003, OR 1.447; 95% CI 1.007–2.082; *p* = 0.003, OR 1.477; 95% CI 1.028–2.082; *p* = 0.035, OR 1.114; 95% CI 1.045–1.189, *p* = 0.001, OR 0.689; 95% CI 0.490–0.964; *p* = 0.031, OR 1.001; 95% CI 1.000–1.002; *p* = 0.016, OR 0.569; 95% CI 0.332–0.971; *p* = 0.039, respectively). The remaining factors did not significantly affect the length of hospitalization. Interestingly, patients with PLA requiring hospitalization for more than 28 HDs were specifically characterized by older age, male sex, a low albumin level and infection with unidentified pathogens compared to those requiring hospitalization for longer than 14 HDs.

## 4. Discussion

The risk factors for a poor prognosis of patients with PLA are well-known, including multiple abscesses, associated malignancies, jaundice, hypoalbuminemia, leukocytosis, bacteremia and *K. pneumoniae* [12,14,28,29,30]. However, there is a need for simple, inexpensive and rapid tests for the prognostic biomarkers presenting in patients with PLA. The PWR is a biomarker that satisfies all these requirements, and its usefulness as a biomarker has been identified in several recent studies [19,20,21,22,23,24,25,26]. We documented the efficacy of PWR as a biomarker in patients with PLA for the first time. In this study, patients with PLA presenting with a low PWR had significantly higher rates of complications, such as metastatic infection and pleural effusion, than patients with a high PWR. In particular, a low PWR was revealed as only one risk factor for the development of metastatic infection in patients with PLA. Interestingly, although *K. pneumoniae* infection is well known as a risk factor for the development of metastatic infection, we did not observe any statistically significant association based on the multivariate analysis. Thus, a lower PWR state renders the patient susceptible to the spread of bacteria. Based on this result, we might recommend testing for metastatic infection earlier in patients with PLA presenting with a low PWR.

In addition, pleural effusion, which develops in approximately 20-50% of patients with PLA, causes considerable distress to patients during treatment [31,32,33,34,35]. The pathophysiology of the development of pleural effusion in patients with PLA is known to result from inflammation of the diaphragm and increased permeability of the lymphatics [36]. Although patients with PLA suffer from pleural effusion during the treatment period, few reports have described analyses of the risk factors related to pleural effusion. In the present study, 10.2% of the total number of patients with PLA developed pleural effusion, and a low PWR, ALP level and infection with an unidentified organism were identified as risk factors for the development of pleural effusion. Although further studies are required, periodic chest X-ray examinations are recommended for patients with PLA presenting with a low PWR because of the high possibility of pleural effusion.

According to previous studies, the cutoff value for PWR is 20 in patients who undergo splenectomy for trauma, as a reliable marker of infection [25], and 26 in patients with ovarian cancer [20], as a risk factor for a poor prognosis. In our study, a PWR less than 17.05 indicated a poor condition. The complication groups had a lower PWR, forming a consensus with the literature that a lower PWR indicates an unstable homeostatic immune system [37]. A low PWR was correlated with older age, a higher hemoglobin level, higher ALT level, lower albumin level, increased CRP level and infections with unidentified causative pathogens. Although *K. pneumoniae* was not significantly associated with a low PWR in the multivariate analysis, *K. pneumoniae* was a major causative pathogen in the low PWR group (56.3%) compared to the high PWR group (39.8%). The relationship between lower platelet counts and *K. pneumoniae* infection indicates the importance of understanding the role of vascular interactions, especially the role of platelets, in determining the prognoses of diverse medical conditions. Previous studies also noted that thrombocytopenia is a sign of severe infection, which may be explained by diverse cell receptors, such as Toll-like receptors, And its role in activation of neutrophils. The finding that platelets activate the antibacterial mode of the neutrophils and form neutrophil extracellular traps also provides a reason for placing clinical emphasis on thrombocytopenia [37]. Despite a reactive increase in WBC counts, thrombocytopenia significantly hastens immunologic reactions and forms a vicious cycle of organ dysfunction. Therefore, it is important to know the tipping point for this cycle and respond with an appropriate treatment. A low PWR, which involves thrombocytopenia, was frequently observed in patients infected with *K. pneumoniae*. This result is consistent with a previous study, showing that a blood-culture-confirmed bacterial infection and thrombocytopenia were associated with a poor prognosis [38]. The mechanism by which the bacteria affect the platelets is understood to involve adenosine-diphosphate-induced platelet aggregation and apoptosis and the inhibition of megakaryocytes [39]. Despite the need for subsequent studies, including a precise assessment of pathogen serotypes, adequate measures should be developed in order to achieve a better prognosis while maintaining a higher PWR.

PWR was not statistically associated, to a significant extent, with the mortality of patients with PLA, and statistical significance was difficult to determine, as only 34 patients died. However, in the analysis of factors related to the length of the hospital stay, the PWR was revealed as an important risk factor. A higher percentage of patients requiring prolonged hospitalized showed a low PWR after both 14 days and 28 days. Based on these results, the PWR may offer a prognostic factor and evaluation measure for infectious diseases, specifically for patients with PLA. The risk factors for requiring hospitalization for more than 28 days were a low PWR, older age, male sex, size of the abscess, levels of albumin and ALP and infection with unidentified causative pathogens. Albumin is one of the factors used to measure the liver synthesis rate, but it is also affected by malnutrition. Hypoalbuminemia may be caused by inflammation, hepatocyte damage, decreased albumin synthesis, the dietary insufficiency of amino acids or increased excretion of albumin [40]. Hypoalbuminemia can directly affect innate immunity and the antimicrobial defenses. Associations with infectious diseases might also lead to the identification of low albumin levels as a culprit, causally contributing to both the acquisition and development of complications of infections [41]. Interestingly, hospital stays were longer in patients with culture-negative PLA compared to patients with *K. pneumonia* PLA. Although our data did not show a significant difference in the mortality rate between *K. pneumonia* PLA and culture-negative PLA (data were not shown), Chan et al. demonstrated that *K. pneumonia* PLA has a lower mortality rate than non-*K pneumonia* PLA in a systemic review and meta-analysis [42]. However, Shelet et al. reported that *K. pneumonia* PLA and culture-negative PLA have several demographic and clinical differences, and that the overall outcomes of culture-negative PLA patients are similar to those of *K. pneumonia* PLA patients if they receive an appropriate antibiotic treatment and PD [43]. Therefore, the type of causative organism is important, but it is considered that the most essential factor in the prognosis of a patient with PLA is the timely administration of appropriate antibiotics and PD from the start of treatment for PLA. Generally, the mortality rate is higher in immunocompromised and elderly patients. Several studies have demonstrated that patients presenting with bacteremia, septic shock, cirrhosis, renal failure or cancer also have a higher mortality. Multiple abscesses, associated cancer, jaundice, hypoalbuminemia, leukocytosis, bacteremia and any significant complications are associated with increased mortality [7,10,11,14,42]. Therefore, we should not overlook the fact that other clinical factors should be considered important in the prognosis of patients with PLA besides PWR.

This study has some limitations. Firstly, it was a retrospective study, and we were unable to determine the virulence factors, such as the serotypes of *K. pneumoniae*. Secondly, while this study population was relatively large, we were unable to identify a risk factor for mortality due to the low number of deaths (34, 4.1%). Thirdly, we were unable to confirm the baseline PWR level of the included patients before the development of PLA. Additionally, a meaningful PWR cutoff value remains to be determined. Therefore, our cause-and-effect relationship of PWR in patients with PLA may not be coherent, and further prospective multicenter studies are still needed in order to verify the role of PWR. Further prospective studies are needed in order to investigate whether a more aggressive treatment for patients with PLA presenting with a low PWR would be beneficial.

## 5. Conclusions

In conclusion, patients with PLA presenting with a low PWR exhibit worse clinical manifestations, more complications and a poorer prognosis. The PWR level may be used to measure the disease severity and predict the prognosis of patients with PLA. Since the development of metastatic infection and pleural effusion occur at a higher rate in patients with PLA presenting with a low PWR, the regular monitoring of the PWR might be useful for detecting complications earlier and improving the patient’s prognosis. Considering its cost-effectiveness, PWR could be a novel biomarker used to predict complications and a poor prognosis of PLA.

## Figures and Tables

**Figure 1 diagnostics-12-02556-f001:**
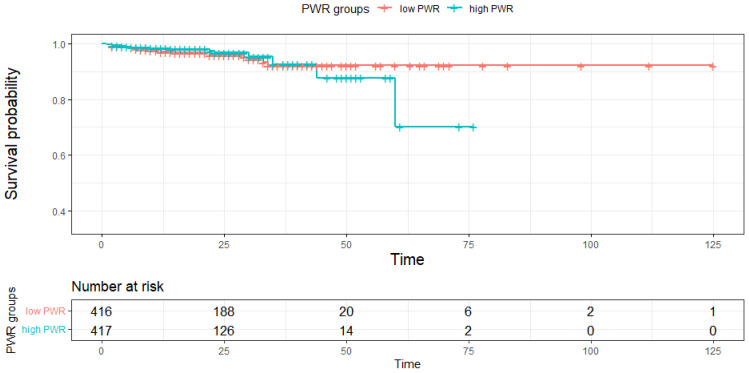
Kaplan–Meier curves of the PWR groups. In a total of 833 patients, a slight decrease in the overall mortality rate was observed, but the difference was not significant between the PWR groups.

**Figure 2 diagnostics-12-02556-f002:**
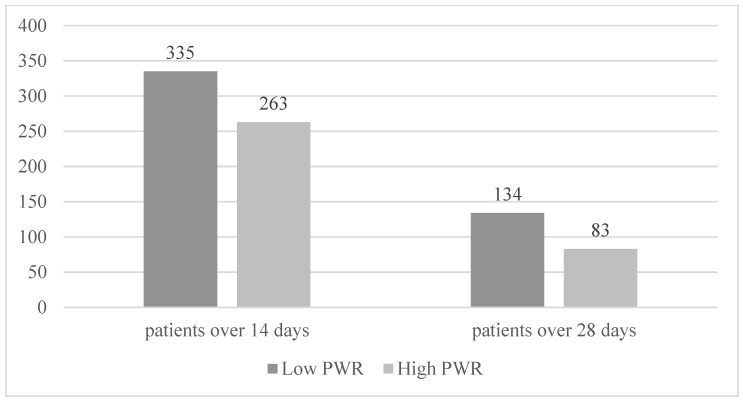
Patient stratification by hospital days. Stratification at hospital days 14 and 28 revealed that the low PWR group required prolonged hospitalization.

**Table 1 diagnostics-12-02556-t001:** Baseline characteristics of patients with liver abscesses according to the platelet counts.

Clinical Characteristics	Total (n = 833)	Low PWR (n = 416)	High PWR (n = 417)	*p* Value
Age (years)	62.2 ± 15.4	64.1 ± 14.9	60.4 ± 15.6	<0.001
Male, n (%)	536 (64.3)	278 (66.8)	258 (61.9)	0.14
BMI, kg/m^2^	23.9 ± 3.4	24.0 ± 3.4	23.8 ± 3.4	0.44
Diabetes mellitus, n (%)	218 (26.2)	107 (25.7)	111 (26.6)	0.77
Hypertension, n (%)	306 (36.7)	168 (40.4)	138 (33.1)	0.03
Laboratory finding				
Anemia, n (%)	313 (37.6)	134 (32.2)	179 (42.9)	0.001
WBC (×10^3^)	13.0 ± 6.5	15.4 ± 7.0	10.5 ± 5.0	<0.001
Platelet count (×10^3^)	234.7 ± 14.1	157.6 ± 80.7	311.6 ± 146.5	<0.001
AST, IU/L	105.4 ± 311.6	126.5 ± 394.7	84.4 ± 194.5	0.051
ALT, IU/L	86.0 ± 121.7	96.0 ± 115.9	76.0 ± 126.5	0.02
ALP, IU/L	249.0 ± 239.5	237.6 ± 229.4	260.4 ± 249.0	0.17
Albumin, g/dL	3.4 ± 0.6	3.3 ± 0.6	3.5 ± 0.6	<0.001
PT-INR	1.3 ± 0.1	1.4 ± 2.3	1.3 ± 0.7	0.33
Cr, mg/dL	1.4 ± 4.1	1.5 ± 4.2	1.2 ± 3.8	0.18
CRP, mg/dL	184.4 ± 250.1	206.8 ± 91.3	161.9 ± 340.7	0.01
Abscess size, cm	5.4 ± 2.8	5.4 ± 2.6	5.3 ± 3.1	0.71
Abscess location, n (%)				
Right	527 (63.3)	248 (59.6)	279 (66.9)	0.03
Left	172 (20.6)	100 (24.0)	72 (17.3)	0.02
Both	134 (16.1)	68 (16.3)	66 (15.8)	0.84
Complications, n (%)	181 (21.7)	124 (29.8)	57 (13.7)	<0.001
Metastatic infection	88 (48.6)	63 (50.8)	25 (43.9)	0.0014
Septic endophthalmitis	15 (8.3)	12 (9.7)	3 (5.3)	0.02
Pneumonia	29 (16.0)	22 (17.7)	7 (12.3)	0.004
Other metastatic infection	44 (24.3)	29 (23.4)	15 (26.3)	0.0019
Abscess rupture	8 (4.4)	5 (4.0)	3 (5.3)	0.48
Pleural effusion	85 (47.0)	56 (45.2)	29 (50.9)	0.0019
Causative organisms, n (%)				
*K. pneumoniae*	400 (48.0)	234 (56.3)	166 (39.8)	<0.001
*E. coli*	59 (7.1)	34 (8.2)	25 (6.0)	0.22
Others	163 (19.6)	27 (6.5)	31 (7.4)	<0.001
Not identified	211 (25.3)	121 (29.1)	195 (46.8)	<0.001

ALP: alkaline phosphatase; ALT: alanine aminotransferase; AST: aspartate aminotransferase; BMI: body mass index; Cr: creatinine; CRP: C-reactive protein; PT-INR: prothrombin time in international normalized ratio; WBC: white blood cell.

**Table 2 diagnostics-12-02556-t002:** Univariate and multivariate analyses of factors related to a low PWR in patients with PLA.

Variable	Univariate	*p* Value	Multivariate	*p* Value
	OR (95% CI)		OR (95% CI)	
Older age (≥65 years)	1.567 (1.192–2.061)	0.0013	1.918 (1.330–2.780)	<0.001
Male	0.805 (0.606–1.070)	0.136		
Hypertension	1.370 (1.033–1.819)	0.029	1.224 (0.857–1.750)	0.266
DM	0.955 (0.701–1.300)	0.768		
Location of the abscess (Right)	1.646 (0.975–2.830)	0.065		
Abscess size, cm	1.009 (0.962–1.059)	0.710		
Anemia	0.634 (0.477–0.841)	<0.001	0.423 (0.288–0.615)	<0.001
Albumin, g/dL	0.575 (0.449–0.731)	<0.001	0.617 (0.447–0.844)	<0.001
AST, IU/L	1.001 (1.0001–1.002)	<0.001	0.9998(0.998–1.002)	0.820
ALT, IU/L	1.0016 (1.0003–1.003)	0.024	1.0007 (0.998–1.004)	0.657
ALP, IU/L	0.9995(0.9990–1.0001)	0.170		
PT-INR	1.057 (0.961–1.307)	0.412		
Cr, mg/dL	1.034 (0.991–1.131)	0.274		
CRP, mg/dL	1.003 (1.002–1.004)	<0.001	1.0005 (0.999–1.002)	<0.001
Causative organisms				
*K. pneumoniae*	1.944 (1.477–2.564)	<0.001	0.829 (0.531–1.291)	0.410
*E. coli*	1.396 (0.820–2.405)	0.222		
Others	1.093 (0.706–1.696)	0.690		
Not identified	0.406 (0.291–0.565)	<0.001	0.545 (0.334–0.883)	0.014

OR: odds ratio; CI: confidence interval; ALP: alkaline phosphatase; ALT: alanine aminotransferase; AST: aspartate aminotransferase; Cr: creatinine; CRP: C-reactive protein; DM: diabetes mellitus; PT-INR: prothrombin time test in international normalized ratio; WBC: white blood cell. Data are presented as odds ratios (95% confidence intervals).

**Table 3 diagnostics-12-02556-t003:** Risk factors for metastatic infection in patients with pyogenic liver abscesses.

Variable	Univariate	*p* Value	Multivariate	*p* Value
	OR (95% CI)		OR (95% CI)	
Low PWR (<17.05)	2.591 (1.607–4.293)	<0.001	1.953 (1.163–3.364)	0.013
Older age (≥65 years)	1.425 (0.907–2.255)	0.126		
Hypertension	1.192 (0.747–1.883)	0.454		
Diabetes mellitus	0.934 (0.544–1.546)	0.797		
Abscess location (Right)	0.839 (0.531–1.338)	0.454		
Abscess size, cm	1.017 (0.939–1.098)	0.672		
Anemia	0.521 (0.304–0.857)	0.013	0.609 (0.339–1.052)	0.084
Albumin, g/dL	0.719 (0.489–1.056)	0.093		
AST, IU/L	1.000 (0.999–1.0006)	0.764		
ALT, IU/L	1.000 (0.999–1.002)	0.467		
ALP, IU/L	1.000 (0.999–1.001)	0.826		
PT-INR	0.938 (0.467–1.090)	0.722		
Cr, mg/dL	0.994 (0.868–1.039)	0.854		
CRP, mg/dL	1.000 (0.9997–1.001)	0.21		
Causative organisms				
*K. pneumoniae*	2.221 (1.396–3.601)	0.0009	1.304 (0.688–2.625)	0.433
*E. coli*	0.296 (0.048–0.974)	0.095		
Others	1.183 (0.572–2.245)	0.627		
Not identified	0.459 (0.231–0.843)	0.018	0.648 (0.279–1.496)	0.307

OR: odds ratio; CI: confidence interval; ALP: alkaline phosphatase; ALT: alanine aminotransferase; AST: aspartate aminotransferase; Cr: creatinine; CRP: C-reactive protein; PT-INR: prothrombin time in international normalized ratio.

**Table 4 diagnostics-12-02556-t004:** Risk factors for pleural effusion in patients with pyogenic liver abscesses.

Variable	Univariate	*p* Value	Multivariate	*p* Value
	OR (95% CI)		OR (95% CI)	
Low PWR (<17.05)	2.081 (1.310–3.371)	0.002	2.007 (1.204–3.419)	0.0087
Older age (≥65 years)	1.121 (0.715–1.759)	0.617		
Hypertension	0.881 (0.543–1.401)	0.598		
Diabetes mellitus	0.794 (0.454–1.331)	0.399		
Abscess location (Right)	1.641(1.008–2.757)	0.527		
Abscess size, cm	1.141 (1.060–1.227)	<0.001	1.077 (0.986–1.173)	0.095
Anemia	1.627 (1.034–2.555)	0.034	1.237 (0.733–2.075)	0.422
Albumin, g/dL	0.462 (0.312–0.679)	<0.001	0.736 (0.461–1.175)	0.198
AST, IU/L	1.0001 (0.9993–1.0006)	0.613		
ALT, IU/L	1.0008 (0.9992–1.002)	0.251		
ALP, IU/L	1.001 (1.0005–1.002)	<0.001	1.001 (1.000–1.002)	0.006
PT-INR	1.017 (0.838–1.113)	0.751		
Cr, mg/dL	0.937 (0.690–1.031)	0.571		
CRP, mg/dL	1.0001 (0.998–1.0008)	0.679		
Causative organisms				
*K. pneumoniae*	1.459 (0.931–2.305)	0.101		
*E. coli*	2.168 (1.031–4.214)	0.030	1.362 (0.592–2.906)	0.443
Others	1.538 (0.799–2.787)	0.174		
Not identified	0.345 (0.167–0.640)	0.002	0.458 (0.218–0.883)	0.027

OR: odds ratio; CI: confidence interval; ALP: alkaline phosphatase; ALT: alanine aminotransferase; AST: aspartate aminotransferase; Cr: creatinine; CRP: C-reactive protein; PT-INR: prothrombin time in international normalized ratio.

**Table 5 diagnostics-12-02556-t005:** Risk factors for prolonged hospitalization for more than 14 days in patients with pyogenic liver abscesses.

Variable	Univariate	*p* Value	Multivariate	*p* Value
	OR (95% CI)		OR (95% CI)	
Low PWR (<17.05)	2.422 (1.774–3.325)	<0.001	1.573 (1.053–2.357)	0.027
Older age (≥65 years)	1.265 (0.934–1.715)	0.13		
Male	1.226 (0.893–1.694)	0.211		
Hypertension	1.147 (0.838–1.577)	0.395		
Diabetes mellitus	1.309 (0.923–1.877)	0.137		
Abscess location (Right)	0.850 (0.617–1.163)	0.313		
Abscess size, cm	1.253 (1.176–1.339)	<0.001	1.094 (1.011–1.186)	0.028
Anemia	1.452 (1.057–2.008)	0.022	1.285 (0.826–1.971)	0.254
Albumin, g/dL	0.457 (0.343–0.602)	<0.001	0.788 (0.547–1.125)	0.195
AST, IU/L	0.9997 (0.9991–1.0002)	0.279		
ALT, IU/L	1.0007 (0.9993–1.002)	0.369		
ALP, IU/L	1.002 (1.001–1.003)	<0.001	1.002 (1.0007–1.003)	0.002
PT-INR	1.020 (0.933–1.245)	0.724		
Cr, mg/dL	1.033 (0.982–1.175)	0.44		
CRP, mg/dL	1.008 (1.007–1.010)	<0.001	1.005 (1.003–1.008)	<0.001
Causative organisms				
*K. pneumoniae*	2.678 (1.951-3.702)	<0.001	1.519 (0.885–2.578)	0.124
*E. coli*	1.423 (0.776-2.791)	0.276		
Others	1.041 (0.642–1.738)	0.874		
Not identified	0.287 (0.202–0.408)	<0.001	0.604 (0.351–1.026)	0.064

OR: odds ratio; CI: confidence interval; ALP: alkaline phosphatase; ALT: alanine aminotransferase; AST: aspartate aminotransferase; Cr: creatinine; CRP: C-reactive protein; PT-INR: prothrombin time in international normalized ratio.

**Table 6 diagnostics-12-02556-t006:** Risk factors for prolonged hospitalization for more than 28 days in patients with pyogenic liver abscesses.

Variable	Univariate	*p* Value	Multivariate	*p* Value
	OR (95% CI)		OR (95% CI)	
Low PWR (<17.05)	1. 912 (1.396–2.631)	<0.001	1.738 (1.211–2.507)	0.003
Elderly age (≥65 years)	1.561 (1.144–2.135)	0.005	1.447 (1.007–2.082)	0.003
Male	1.400 (1.018–1.922)	0.038	1.477 (1.028–2.082)	0.035
Hypertension	1.151 (0.835–1.580)	0.387		
Diabetes mellitus	1.375 (0.974–1.931)	0.067		
Abscess location (Right)	0.992 (0.721–1.371)	0.963		
Abscess size, cm	1.163 (1.101–1.229)	<0.001	1.114 (1.045–1.189)	0.001
Anemia	1.226 (0.892–1.682)	0.207		
Albumin, g/dL	0.437 (0.328–0.578)	<0.001	0.689 (0.490–0.964)	0.031
AST. IU/L	1.0001 (0.999–1.0007)	0.463		
ALT, IU/L	1.001 (1.000–1.003)	0.043	1.001 (0.999–1.003)	0.061
ALP, IU/L	1.001 (1.0006–1.002)	<0.001	1.001 (1.000–1.002)	0.016
PT-INR	0.942 (0.661–1.057)	0.57		
Cr, mg/dL	0.997 (0.939–1.033)	0.877		
CRP, mg/dL	1.0009 (0.999–1.002)	0.235		
Causative organisms				
*K. pneumoniae*	1.643 (1.203–2.248)	0.002	1.086 (0.698–1.703)	0.717
*E. coli*	1.630 (0.919–2.818)	0.086		
Others	1.263 (0.778–2.007)	0.333		
Not identified	0.403 (0.264–0.601)	<0.001	0.569 (0.332–0.971)	0.039

OR: odds ratio; CI: confidence interval; ALP: alkaline phosphatase; ALT: alanine aminotransferase; AST: aspartate aminotransferase; Cr: creatinine; CRP: C-reactive protein; PT-INR: prothrombin time in international normalized ratio.

## Data Availability

The material described is not under publication or consideration for publication elsewhere.

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
