# Peer review of "Platelet-to-White Blood Cell Ratio: A Feasible Biomarker for Pyogenic Liver Abscess"

_diagnostics, 2022, doi:10.3390/diagnostics12102556_

Round 1

Reviewer 1 Report

This study is a retrospective study focused on the use of the PWR as a prognostic marker for pyogenic liver abscess, which has certain positive significance for the clinical treatment of liver abscess. And I would like to provide a few personal views on this study for the authors.

1. In the introduction(P1,39), there seems to be a grammatical error here.

2. In different hospitals., patient discharge criteria may not be entirely consistent。Therefore, it is hoped that the author can add the discharge criteria of patients in the article, which will undoubtedly affect the length of stay.

3. In the Discussion(P11,290), the authors state that the low PWR group included significantly more patients having abscesses on both sides of the liver. In Table1,occurrence in both lobes was not different between the PWR groups (16.3% vs. 15.8%, 151 p=0.84). Please clarify.

Author Response

Reviewer 1

This study is a retrospective study focused on the use of the PWR as a prognostic marker for pyogenic liver abscess, which has certain positive significance for the clinical treatment of liver abscess. And I would like to provide a few personal views on this study for the authors.

  1. In the introduction (P1,39), there seems to be a grammatical error here.

Answer: Thank you for your comment. We correct the grammatical error. Pyogenic liver abscess (PLA) has low and varying -> Pyogenic liver abscess (PLA) has low and various

  1. In different hospitals, patient discharge criteria may not be entirely consistent. Therefore, it is hoped that the author can add the discharge criteria of patients in the article, which will undoubtedly affect the length of stay.
    Answer: Thank you for your valuable comment. According to your comment, we newly added the discharge criteria in 2.2 treatment protocol section (page 3).

Discharge criteria among patients who had received intravenous antibiotic treatment around 2 weeks were defined as when 1) the patient’s vital signs were stable, 2) the inflammatory markers such as WBC count and CRP level were improved, and 3) the drain could be removed.

  1. In the Discussion (P11, 290), the authors state that the low PWR group included significantly more patients having abscesses on both sides of the liver. In Table1,occurrence in both lobes was not different between the PWR groups (16.3% vs. 15.8%, 151 p=0.84). Please clarify.

Answer: Thank for your detailed comment. Your comment is correct. Therefore, we removed the sentence “The location of the abscess was also another key factor. PLA was mostly located on the right side, considering the location of the liver. However, the low PWR group included significantly more patients having abscesses on both sides of the liver.”

Reviewer 2 Report

Dear Authors, I congratulate you for writing a large case series on PLA from multiple hospitals over a long time and with about 4% mortality, which is commendable. I read your paper with interest, and I agree that PWR is novel in PLA. I also feel that though this is novel, this is actually intuitive that patients with high white cells and low platelets or either one have worse outcomes, and thus combining makes sense but is not that out of the box kind of thought. I have some comments and hopefully you can address these issues in your manuscript.

1. We are aware of sepsis guidelines, and we also know SIRS criteria and it is common knowledge that Tw <4 and >12 is considered bad. So how do you factor this differential nature of Tw into your ratio? Your ratio seems to consider that higher Tw is bad (as higher Tw gives a lower PWR). But someone with a Tw of 2 will be worse than Tw of 10, and we know that neutropenia is worse in all patients and more so in patients with immune suppression. Is this something you could look into your data and re-analyse? 

2. You find that patients with low PWR had high hemoglobin. Than you found on multivariate analysis that low PWR patients had anemia. This is paradoxical. Please check the data. Pls explain why is there a discrepancy?

3. Your average PLA size is more than 5.4cm. I do not read in the entire manuscript about the drainage policy and management care bundle for PLA (PMID - 26133908).

4. Your data shows that patients with culture negative PLA is worse than patients with Klebsiella PLA. However our data from Singapore showed that both have similar outcomes if we abide with empirical abx and drainage policy and care bundles. This is interesting to me and would like authors to put down their views in the discussion section on this. (PMID - 27733320). 

5. The length of stay of 14 and 28 days, while a good measure, it is actually multifactorial and thus, relying on PWR is not nice. I agree it is simple, cheap and easy to measure. But LOS is not determined by PWR. I feel the underlaying patient condition and ongoing clinical management both determines LOS and PWR. With this retrospective data just because we got p value significance, we should not conclude it as such. Pls moderate this in discussion. LOS would be determined by comorbidity, organ failure and management, drainage, duration of IV antibiotics, other issues etc. PWR is a bystander effect and not a direct contributor to LOS. Agree it can give some insight to the treating doctors. 

6. I did a recent metaanalysis and reported that Klebsiella PLA has lower mortality than non-Klebsiella PLA. This was a bit surprising to me, but it is the existing data out there. You have a substantial Klebsiella PLA. Could this be the reason for your lower mortality statistic? (PMID - 36145408)

Thanks

Author Response

Reviewer 2

Dear Authors, I congratulate you for writing a large case series on PLA from multiple hospitals over a long time and with about 4% mortality, which is commendable. I read your paper with interest, and I agree that PWR is novel in PLA. I also feel that though this is novel, this is actually intuitive that patients with high white cells and low platelets or either one have worse outcomes, and thus combining makes sense but is not that out of the box kind of thought. I have some comments and hopefully you can address these issues in your manuscript.

We deeply appreciated to the reviewer for their in-depth and important comments.

  1. We are aware of sepsis guidelines, and we also know SIRS criteria and it is common knowledge that Tw <4 and >12 is considered bad. So how do you factor this differential nature of Tw into your ratio? Your ratio seems to consider that higher Tw is bad (as higher Tw gives a lower PWR). But someone with a Tw of 2 will be worse than Tw of 10, and we know that neutropenia is worse in all patients and more so in patients with immune suppression. Is this something you could look into your data and re-analyse? 

Answer: Thank for your detailed comment. Although the correlation between the criteria presented by SIRS and the PWR level is quite high and it might tell about the prognosis in the PLA patient group with TW<4 or >12, I think that PWR offers a different clinical meaning.

Clinical characteristics

Total (n=833)

Low PWR (n=416)

High PWR (n=417)

p value

TW<4 or >12, n (%)

462 (55.2)

275 (66.1)

187 (40.5)

<0.001

  1. You find that patients with low PWR had high hemoglobin. Then you found on multivariate analysis that low PWR patients had anemia. This is paradoxical. Please check the data. Pls explain why is there a discrepancy?

Answer: Thank you for your valuable comment. When Hemoglobin (Hb) was set as a continuous variable, Hb levels were slightly higher in patients with low PWR, but when analyzed based on anemia, 32.2% of patients with low PWR and 42.9% of patients with high PWR showed anemia. This is thought to be a phenomenon seen when the continuous data is transformed into a dichotomous variable. However, as the reviewer pointed out, it was judged that the reader had difficulties in interpretation, so the table in Table 1 was changed from Hb level to anemia.

Table 1

Anemia, n (%)

313 (37.6)

134 (32.2)

179 (42.9)

0.001

  1. Your average PLA size is more than 5.4cm. I do not read in the entire manuscript about the drainage policy and management care bundle for PLA (PMID - 26133908).

Answer: Thank for your valuable comment. According to your comment, we added additional section as 2.2 treatment protocol in material and methods.

The basis of treatment was antibiotic therapy and percutaneous drainage (PD) according to size and nature of PLA. All patients with PLA treated promptly when PLA is suspected after at least one set of blood culture taken. Empiric antibiotic regimens usually consist of a third-generation cephalosporin, such as ceftriaxone, along with metronidazole. PD would be performed when (1) size of PLA ≥ 5 cm (2) presence of hemodynamic instability (3) gas within the abscess cavity regardless of size and (4) failure of antibiotic therapy for PLA < 5cm (5) sign of impending rupture on radiologic imaging. Antibiotic regimen was subsequently tailored to culture and sensitivity results of blood or pus. The duration of intravenous antibiotic treatment is recommended around 2 weeks before transitioning to oral antibiotics. Total duration of antibiotic treatment varied according to the clinical and radiological response for each PLA patient. Depending on the radiologic assessment of size and status of residual PLA; the drainage tube was either repositioned, upsized or withdrawn. The drains were removed when decrease <10 mL/day drain aspirates for at least 2 days or resolution of PLA with clinical improvement as evidenced by stable vital sign, WBC and C-reactive protein (CRP) down trending. Discharge criteria among patients who had received intravenous antibiotic treatment around 2 weeks were defined as when 1) the patient’s vital signs were stable, 2) the inflammatory markers such as WBC count and CRP level were improved, and 3) the drain could be removed. Treatment outcomes were measured by length of in hospital stay and mortality. The length of hospitalization was 14 days and 28 days, respectively. All the patients recommended a repeat radiologic imaging at 2–3 weeks intervals to evaluate treatment response. Antibiotic treatment was discontinued when there was a clinical resolution and near complete to complete radiological resolution of PLA, after a minimum duration of 6 weeks.

  1. Your data shows that patients with culture negative PLA is worse than patients with Klebsiella PLA. However, our data from Singapore showed that both have similar outcomes if we abide with empirical abx and drainage policy and care bundles. This is interesting to me and would like authors to put down their views in the discussion section on this. (PMID - 27733320).

 Answer: Thank for your valuable comment. According to your comment, we added the view in discussion section (page13)

Interestingly, hospital stays were longer in patients with culture negative PLA compare to patients with K. pneumonia PLA. Although our data did not show significant difference of mortality rate between K. pneumonia PLA and culture negative PLA (data was not shown), Chan et al demonstrated that K. pneumonia PLA has lower mortality rate than non-K pneumonia PLA in a systemic review and meta-analysis (PMID 36145408). However, Shelet et al. reported K.pneumonia PLA and culture negative PLA have several demographic and clinical differences, the overall outcomes of culture negative PLA patients are similar to K. pneumonia PLA patients if they have an appropriate antibiotic treatment and PD (PMID 2773320). Therefore, the type of causative organisms is important, but it is considered that the most essential factor in the prognosis of a patient with PLA is that appropriate antibiotics and PD are performed quickly from the start of treatment of PLA.

  1. The length of stay of 14 and 28 days, while a good measure, it is actually multifactorial and thus, relying on PWR is not nice. I agree it is simple, cheap and easy to measure. But LOS is not determined by PWR. I feel the underlaying patient condition and ongoing clinical management both determines LOS and PWR. With this retrospective data just because we got p value significance, we should not conclude it as such. Pls moderate this in discussion. LOS would be determined by comorbidity, organ failure and management, drainage, duration of IV antibiotics, other issues etc. PWR is a bystander effect and not a direct contributor to LOS. Agree it can give some insight to the treating doctors. 

Answer: Thank for your valuable comment. I totally agree with your opinion. I also agree with your concerns. Therefore, the following sentence has been inserted into the discussion (Page 13).

Generally, mortality rate is higher in immunocompromised and elderly patients. Several studies have demonstrated that patients presenting with bacteremia, septic shock, cirrhosis, renal failure, or cancer also have a higher mortality. Multiple abscesses, associated cancer, jaundice, hypoalbuminemia, leukocytosis, bacteremia, and any significant complications are associated with increased mortality. Therefore, it should not be overlooked that other clinical factors should be considered important in the prognosis of patients with PLA besides PWR.

  1. I did a recent meta-analysis and reported that Klebsiella PLA has lower mortality than non-Klebsiella PLA. This was a bit surprising to me, but it is the existing data out there. You have a substantial Klebsiella PLA. Could this be the reason for your lower mortality statistic? (PMID - 36145408)

Answer: Thank for your detailed comment. The mortality of K. pneumonia PLA patients (3.0%) was slightly lower than other organism (5.1%). However, it is not statistically significant. Therefore, it seems difficult to explain our mortality results in any one way.

Reviewer 3 Report

The authors could include in the introduction other ratios between blood elements that may have predictive value. Some values can be predictive for certain complications, for example neutrophil lymphocyte ratio for anastomotic fistula (Radulescu D, Baleanu VD, Padureanu V, et al. Neutrophil/Lymphocyte Ratio as Predictor of Anastomotic Leak after Gastric Cancer Surgery. Diagnostics (Basel). 2020 Oct 9;10(10):799. doi: 10.3390/diagnostics10100799)

Author Response

Reviewer 3

The authors could include in the introduction other ratios between blood elements that may have predictive value. Some values can be predictive for certain complications, for example neutrophil lymphocyte ratio for anastomotic fistula (Radulescu D, Baleanu VD, Padureanu V, et al. Neutrophil/Lymphocyte Ratio as Predictor of Anastomotic Leak after Gastric Cancer Surgery. Diagnostics (Basel). 2020 Oct 9;10(10):799. doi: 10.3390/diagnostics10100799)

Answer: Thank for your comment. According to your comment, we added the views in introduction section (page 2).

Recently, hematological markers for the prognosis in several diseases have become an area of research focus. The neutrophil-to-lymphocyte ratio, monocyte-to-lymphocyte ratio, red cell distribution width, and mean platelet volume have been suggested to be inflammatory markers, and their prognostic value in various diseases.
